# *3DB*: A Framework for Debugging Computer Vision Models

**Guillaume Leclerc**[†]
LECLERC@MIT.EDU
MIT *

**Hadi Salman**[†]
HADY@MIT.EDU
MIT *

**Andrew Ilyas**[†]
AILYAS@MIT.EDU
MIT

**Sai Vemprala**
SAIHV@MICROSOFT.COM
Microsoft Research

**Logan Engstrom**
ENGSTROM@MIT.EDU
MIT

**Vibhav Vineet**
VIVINEET@MICROSOFT.COM
Microsoft Research

**Kai Xiao**
KAIX@MIT.EDU
MIT

**Pengchuan Zhang**
PENZHAN@MICROSOFT.COM
Microsoft Research

**Shibani Santurkar**
SHIBANI@MIT.EDU
MIT

**Greg Yang**
GE.YANG@MICROSOFT.COM
Microsoft Research

**Ashish Kapoor**
AKAPOOR@MICROSOFT.COM
Microsoft Research

**Aleksander Mądry**
MADRY@MIT.EDU
MIT

## Abstract

We introduce *3DB*: an extendable, unified framework for testing and debugging vision models using photorealistic simulation. We demonstrate, through a wide range of use cases, that *3DB* allows users to discover vulnerabilities in computer vision systems and gain insights into how models make decisions. *3DB* captures and generalizes many robustness analyses from prior work, and enables one to study their interplay. Finally, we find that the insights generated by the system transfer to the physical world. We are releasing *3DB* as a library[1] alongside a set of examples[2], guides[3], and documentation[4].

## 1  Introduction

Modern machine learning models turn out to be remarkably brittle under distribution shift. Indeed, in the context of computer vision, models exhibit an abnormal sensitivity to slight input rotations and translations [18, 37], synthetic image corruptions [32, 38], and changes to the data collection pipeline [49, 19]. Still, while brittleness is widespread, it is often hard to understand its root causes, or even to characterize the precise situations in which this behavior arises.

How do we then comprehensively diagnose model failure modes? Stakes are often too high to simply deploy models and collect "real-world" failure cases. There has thus been a line of work in computer vision focused on identifying systematic sources of model failure such as unfamiliar

---

[*]Work partially completed while at Microsoft Research.
[†]Equal contribution.
[1]https://github.com/3db/3db
[2]https://github.com/3db/blog_demo
[3]https://3db.github.io/3db/usage/quickstart.html
[4]https://3db.github.io/3db/

36th Conference on Neural Information Processing Systems (NeurIPS 2022).

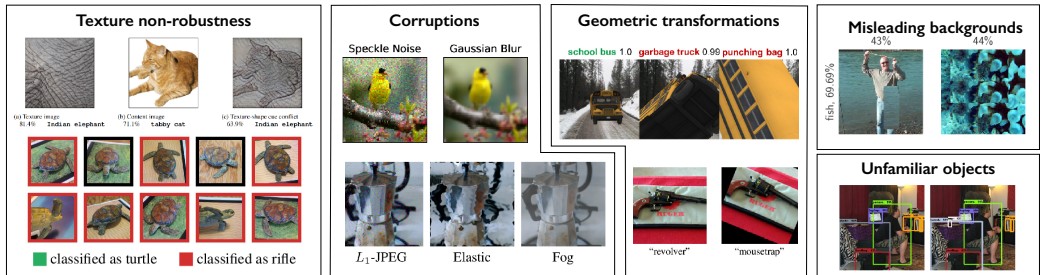

Figure 1: Examples of vulnerabilities of computer vision systems identified through prior in-depth robustness studies. Figures reproduced from [25, 5, 32, 38, 3, 18, 69, 52].

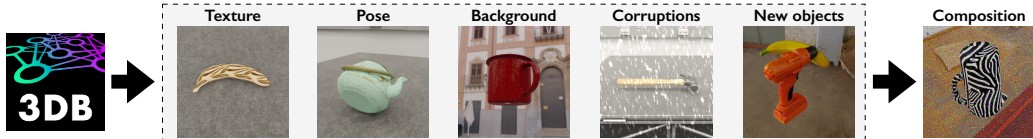

Figure 2: The *3DB* framework is modular enough to facilitate—among other tasks—efficient rediscovery of all the types of brittleness shown in Figure 1. It also allows users to realistically compose transformations (right) while still being able to disentangle the results.

object orientations [3], misleading backgrounds [74, 69], or shape-texture conflicts [25, 5]. These analyses—a selection of which is visualized in Figure 1—reveal patterns or situations that degrade performance of vision models, providing invaluable insights into model robustness. Still, carrying out each such analysis requires its own set of (often complex) tools, usually accompanied by a significant amount of manual labor (e.g., image editing, style transfer), expertise, and data cleaning. This prompts the question:

*Can we support reliable discovery of model failures in a systematic, automated, and unified way?*

**Contributions.** In this work, we propose *3DB*, a framework for automatically identifying and analyzing the failure modes of computer vision models. This framework makes use of a 3D simulator to render realistic scenes that can be fed into any computer vision system. Users can specify a set of transformations to apply to the scene—such as pose changes, background changes, or camera effects—and can also customize and compose them. The system then performs a guided search, evaluation, and aggregation over these user-specified configurations and presents the user with an interactive, user-friendly summary of the model's performance and vulnerabilities. *3DB* is general enough to enable users to, with minimal effort, re-discover insights from prior work on pose, background, and texture bias (cf. Fig. 2), among others. Further, while prior studies have largely been focused on examining model sensitivities along a single axis, *3DB* allows users to compose various transformations and understand the interplay between them, while still being able to disentangle their individual effects.

The remainder of this paper is structured into the following parts: in Section 2 we discuss the design of *3DB*, including the motivating principles, design goals, and concrete architecture used. We highlight how the implementation of *3DB* allows users to quickly experiment, stress-test, and analyze their vision models. Then, in Section 3 we illustrate the utility of *3DB* through a series of case studies uncovering biases in an ImageNet-pretrained classifier. Finally, we show (in Section 4) that the vulnerabilities uncovered with *3DB* correspond to actual failure modes in the physical world (i.e., they are not specific to simulation).

## 2 Designing *3DB*

The goal of *3DB* is to leverage photorealistic simulation to effectively diagnose failure modes of computer vision models. To this end, the following set of principles guide the design of *3DB*:

**Generality.** *3DB* should support any type of computer vision model (i.e., not necessarily a neural network) trained on any dataset and task (i.e., not necessarily classification). Furthermore, the framework should support diagnosing non-robustness with respect to any parameterizable three-dimensional scene transformation.

**Compositionality.** Corruptions and transformations rarely occur in isolation—*3DB* should allow users to investigate robustness along many different axes simultaneously.

**Physical realism.** The vulnerabilities extracted from *3DB* should correspond to models' behavior in the real (physical) world, and, in particular, not depend on artifacts of the simulation process itself.

**User-friendliness.** *3DB* should be simple to use and should relay insights to the user in an easy-to-understand manner. Even non-experts should be able to look at the result of a *3DB* experiment and easily understand what the weak points of their model are, as well as gain insight into how the model behaves more generally.

**Scalability.** *3DB* should be performant and parallel.

## 2.1   Capabilities and workflow

To achieve the goals articulated above, we design *3DB* modularly, i.e., as a combination of swappable components. This combination allows the user to specify transformations they want to test, search over the space of these transformations, and aggregate the results of this search in a concise way. More specifically, the *3DB* workflow revolves around five steps (visualized in Figure 3):

**Setup.** The user collects one or more 3D meshes that correspond to objects the model is trained to recognize, as well as a set of environments to test against.

**Search space design.** The user defines a *search space* by specifying a set of transformations (which *3DB* calls *controls*) that they expect the computer vision model to be robust to (e.g., rotations, translations, zoom, etc.). Controls are grouped into "rendered controls" (applied during the rendering process) and "post-processor controls" (applied after the rendering as a 2D image transformation).

**Policy-guided search.** After the user has specified a set of controls, *3DB* instantiates and renders a myriad of object configurations derived from compositions of the given transformations. It records the behavior of the ML model on each constructed scene for later analysis. A user-specified *search policy* over the space of all possible combinations of transformations determines the scenes for *3DB* to render.

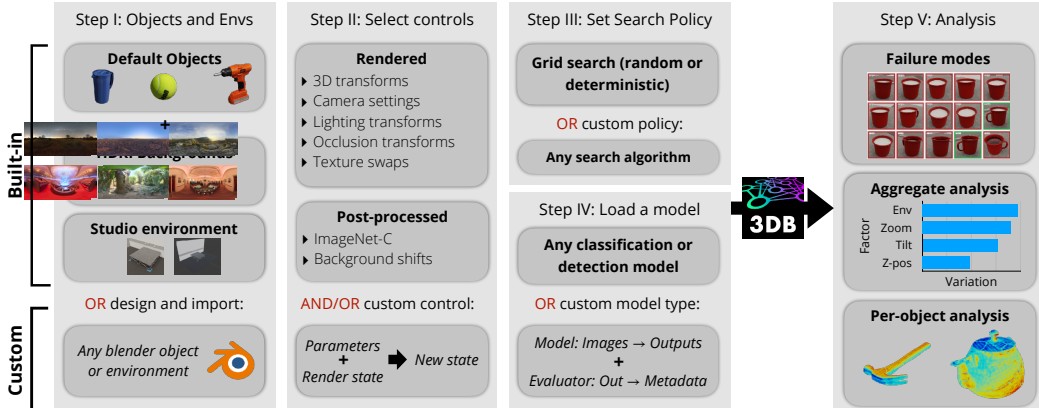

Figure 3: An overview of the *3DB* workflow: First, the user specifies a set of 3D object models and environments to use for debugging. The user also enumerates a set of (in-built or custom) transformations, known as controls, to be applied by *3DB* while rendering the scene. Based on a user-specified search policy over all these controls (and their compositions), *3DB* then selects the exact scenes to render. The computer vision model is finally evaluated on these scenes and the results are logged in a user-friendly manner in a custom dashboard.

**Model loading.** The only remaining step before running a *3DB* analysis is loading the model that the user wants to analyze (e.g., a pre-trained classifier or object detector).

**Analysis and insight extraction.** Finally, *3DB* is equipped with a model *dashboard* (cf. Appendix C) that can read the generated log files and produce a user-friendly visualization of the generated insights. By default, the dashboard has three panels. The first of these is failure mode display, which highlights configurations, scenes, and transformations that caused the model to misbehave. The per-object analysis pane allows the user to inspect the model's performance on a specific 3D mesh (e.g., accuracy, robustness, and vulnerability to groups of transformations). Finally, the aggregate analysis pane extracts insights about the model's performance averaged over all the objects and environments collected and thus allows the user to notice consistent trends and vulnerabilities in their model.

Each of the aforementioned components (the controls, policy, renderer, inference module, and logger) are fully customizable and can be extended or replaced by the user without altering the core code of *3DB*. For example, while *3DB* supports more than 10 types of controls out-of-the-box, users can add custom ones (e.g., geometric transformations) by implementing an abstract function that maps a 3D state and a set of parameters to a new state. Similarly, *3DB* supports debugging classification and object detection models by default, and by implementing a custom evaluator module, users can extend support to a wide variety of other vision tasks and models. We refer to Appendix B for more on *3DB* design principles, implementation, and scalability.

## 3  Debugging and analyzing models with *3DB*

In this section, we illustrate through case studies how to analyze and debug vision models with *3DB*. In each case, we follow the workflow outlined in Section 2.1—importing the relevant objects, selecting the desired transformations (or constructing custom ones), selecting a search policy, and finally analyzing the results.

In all our experiments, we analyze a ResNet-18 [30] trained on the ImageNet [53] classification task (its validation set accuracy is 69.8%). Note that *3DB* is classifier-agnostic (i.e., ResNet-18 can be replaced with any PyTorch classification module), and even supports object detection tasks. For our analysis, we collect 3D models for 16 ImageNet classes (see Appendix F for more details on each experiment). We ensure that in "clean" settings, i.e., when rendered in simple poses on a plain white background, the 3D models are correctly classified at a reasonable rate (cf. Table 1) by our pre-trained ResNet.

Table 1: Accuracy of a pre-trained ResNet-18, for each of the 16 ImageNet classes considered, on the corresponding 3D model we collected, rendered at an unchallenging pose on a white background ("Simulated" row); and the subset of the ImageNet validation set corresponding to the class ("ImageNet" row).

|                        | banana | baseball | bowl | drill | golf ball | hammer | lemon | mug  |
|------------------------|--------|----------|------|-------|-----------|--------|-------|------|
| Simulated accuracy (%) | 96.8   | 100.0    | 17.5 | 63.3  | 95.0      | 65.6   | 100.0 | 13.4 |
| ImageNet accuracy (%)  | 82.0   | 66.0     | 84.0 | 40.0  | 82.0      | 54.0   | 76.0  | 42.0 |

### 3.1  Sensitivity to image backgrounds

We begin our exploration by using *3DB* to confirm ImageNet classifiers' reliance on background signal, as pinpointed by several recent in-depth studies [72, 74, 69]. Out-of-the-box, *3DB* can render 3D models onto HDRI files using image-based lighting; we downloaded 408 such background environments from `hdrihaven.com`. We then used the pre-packaged "camera" and "orientation" controls to render (and evaluate our classifier on) scenes of the pre-collected 3D models at random poses, orientations, and scales on each background. Figure 4 shows random example scenes generated by *3DB* for the "coffee mug" model.

**Analyzing a subset of backgrounds.** In Figure 6, we visualize the performance of a ResNet-18 classifier on the 3D models from 16 different ImageNet classes—in random positions, orientations, and scales—rendered onto 20 of the collected HDRI backgrounds. One can observe that background

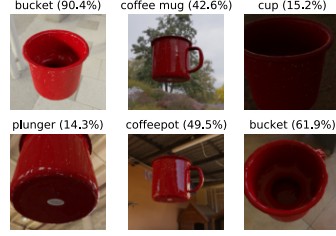

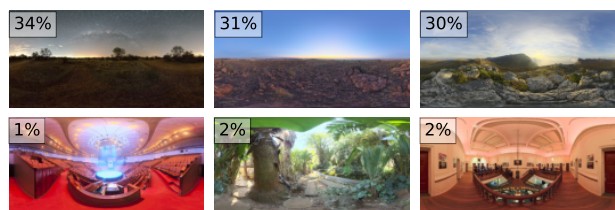

Figure 4: Renderings of the mug 3D model in different environments, labeled with a pretrained model's top prediction.

Figure 5: **(Top)** Best and **(Bottom)** worst background environments for classification of the coffee mug, and their respective accuracies (averaged over camera positions and zoom factors).

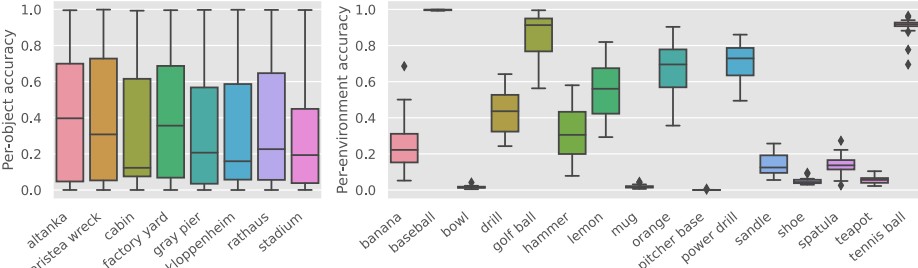

Figure 6: Visualization of accuracy on controls from Section 3.1. **(Left)** We compute the accuracy of the model conditioned on each object-environment pair. For each environment on the x-axis, we plot the variation in accuracy (over the set of possible objects) using a boxplot. We visualize the per-object accuracy spread by including the median line, the first and third quartiles box edges (the interval between which is called the inter-quartile range, IQR), the range, and the outliers (points that are outside the IQR by $3/2|\text{IQR}|$). **(Right)** Using the same format, we track how the classified object (x-axis) impacts variation in accuracy (over different environments) on the y-axis.

dependence indeed varies widely across different objects—for example, the "orange" and "lemon" 3D models depend much more on background than the "tennis ball." We also find that certain backgrounds yield systemically higher or lower accuracy; for example, average accuracy on "gray pier" is five times lower than that of "factory yard."

**Analyzing all backgrounds with the mug model.** The previous study broadly characterizes the classifier's sensitivity to different models and environments. Now, to gain a deeper understanding of this sensitivity, we focus our analysis only a single 3D model (a "coffee mug") rendered in all 408 environments. The highest-accuracy backgrounds had tags such as *skies*, *field*, and *mountain*, while the lowest-accuracy backgrounds had tags *indoor*, *city*, and *building*.

At first, this observation seems to be at odds with the idea that the classifier relies heavily on context clues to make decisions. After all, the backgrounds where the classifier seems to perform well (poorly) are places that we would expect a coffee mug to be rarely (frequently) present in the real world. Visualizing the best and worst backgrounds in terms of accuracy (Figure 5) suggests a possible explanation for this: the best backgrounds tend to be clean and distraction-free. Conversely, complicated backgrounds (e.g., some indoor scenes) often contain context clues that make the mug difficult for models to detect. Comparing a "background complexity" metric (based on the number of edges in the image) to accuracy (Figure 7) supports this explanation: mugs overlaid on more complex backgrounds are more frequently misclassified by the model. In fact, some specific backgrounds even result in the model "hallucinating" objects; for example, the second-most frequent predictions for the *pond* and *sidewalk* backgrounds were *birdhouse* and *traffic light* respectively, despite the fact that neither object is present in the environment.

**Zoom/background interactions case study: the advantage of composable controls.** Finally, we leverage *3DB*'s composability to study interactions between controls. In Figure 8, we plot the mean classification accuracy of our "orange" model while varying background and scale factor. We, for example, find that while the model is highly accurate at classifying "orange" at 2x zoom, the same

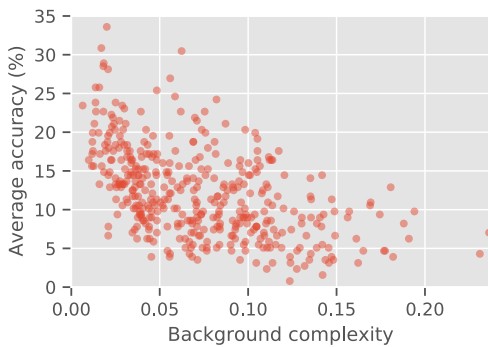

Figure 7: Relation between the complexity of a background and its average accuracy. Here complexity is defined as the average pixel value of the image after applying an edge detection filter.

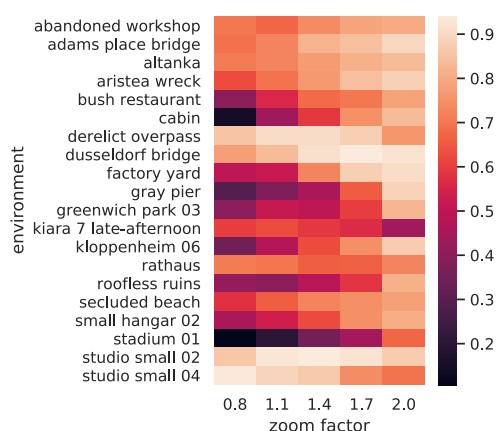

Figure 8: *3DB*'s focus on composability enables us to study robustness along multiple axes simultaneously. Here we study average model accuracy (computed over pose randomization) as a function of *both* zoom level and background.

zoom factor induces failure in a well-lit mountainous environment ("kiara late-afternoon")—a fine-grained failure mode that we would not catch without explicitly capturing the interaction between background choice and zoom.

## 3.2 Texture-shape bias

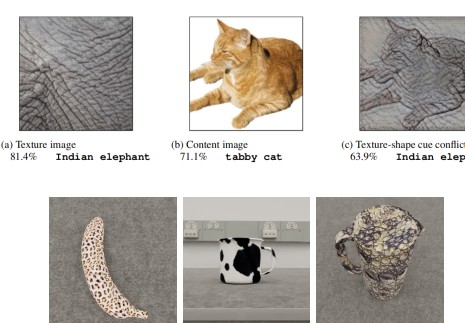

Figure 9: Cue-conflict images generated by Geirhos et al. [25] (*top*) and *3DB* (*bottom*).

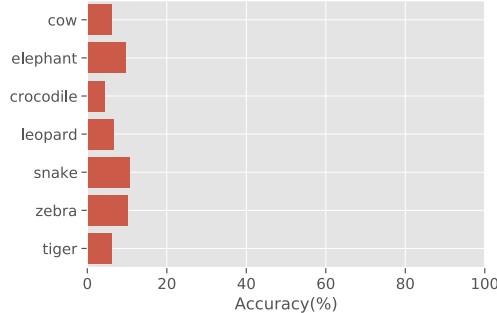

Figure 10: Model accuracy on previously correctly-classified images after their texture is altered via *3DB*, as a function of texture-type.

We now demonstrate how *3DB* can be straightforwardly extended to discover more complex failure modes in computer vision models. Specifically, we will show how to rediscover the "texture bias" exhibited by ImageNet-trained convolutional neural networks (CNNs) [25] in a systematic and (near) photorealistic way. Geirhos et al. [25] fuse pairs of images—combining texture information from one with shape and edge information from the other—to create so-called "cue-conflict" images. They then demonstrate that on these images (cf. Figure 9), ImageNet-trained CNNs typically predict the class corresponding to the texture component, while humans typically predict based on shape.

Cue-conflict images identify a concrete difference between human and CNN decision mechanisms. However, the fused images are unrealistic and can be cumbersome to generate (e.g., even the simplest approach uses style transfer [24]). *3DB* gives us an opportunity to rediscover the influence of texture in a more streamlined fashion.

Specifically, we implement a control (now pre-packaged with *3DB*) that replaces an object's texture with a random (or user-specified) one. We use this control to create cue-conflict objects out of eight

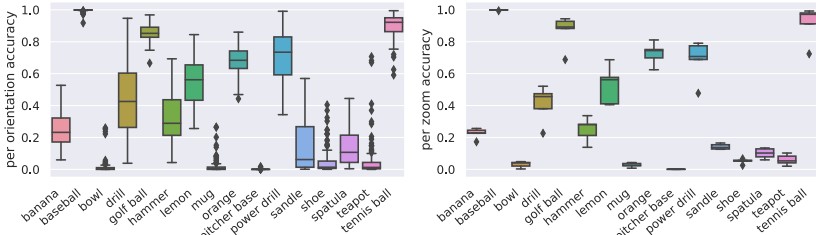

Figure 11: **(Left)** We compute the accuracy of the model for each object-orientation pair. For each object on the x-axis, we plot the variation in accuracy (over the set of possible orientations) using a boxplot. We visualize the per-orientation accuracy spread by including the median line, the first and third quartiles box edges, the range, and the outliers. **(Right)** Using the same format as the left hand plot, we plot how the classified object (on the x-axis) impacts variation in accuracy (over different zoom values) on the y-axis.

3D models[5] and seven animal-skin texture images[6] (i.e., 56 objects in total). We test our pre-trained ResNet-18 on images of these objects rendered in a variety of poses and camera locations. Figure 9 displays sample cue-conflict images generated using *3DB*.

Our study confirms the findings of Geirhos et al. [25] and indicates that texture bias indeed extends to (near-)realistic settings. For images that were originally correctly classified (i.e., when rendered with the original texture), changing the texture reduced accuracy by 90-95% uniformly across textures (Figure 10). Furthermore, we observe that the model predictions usually align better with the texture of the objects rather than their geometry (See Figure 21 in the Appendix).

### 3.3 Orientation and scale dependence

Image classification models are brittle to object orientation in both real and simulated settings [37, 18, 6, 3]. As was the case for both background and texture sensitivity, reproducing and extending such observations is straightforward with *3DB*. Once again, we use the built-in controls to render objects at varying poses, orientations, scales, and environments before stratifying on properties of interest. Indeed, we find that classification accuracy is highly dependent on object orientation (Figure 11 left) and scale (Figure 11 right). However, this dependence is not uniform across objects. As one would expect, the classifier's accuracy is less sensitive to orientation on more symmetric objects (like "tennis ball" or "baseball"), but can vary widely on more uneven objects (like "drill").

For a more fine-grained look at the importance of object orientation, we can measure the classifier accuracy conditioned on a given part of each 3D model being visible. This analysis is once again straightforward in *3DB*, since each rendering is (optionally) accompanied by a UV map which maps pixels in the scene back to locations on on the object surface. Combining these UV maps with accuracy data allows one to construct the "accuracy heatmaps" shown in Figure 12, wherein each part of an object's surface corresponds to classifier accuracy on renderings in which the part is visible. The results confirm that atypical viewpoints adversely impact model performance, and also allow users to draw up a variety of testable hypotheses regarding performance on specific 3D models (e.g., for the coffee mug, the bottom rim is highlighted in red—is it the case that mugs are more accurately classified when viewed from the bottom)? These hypotheses can then be investi-

---

[5]Object models: mug, helmet, hammer, strawberry, teapot, pitcher, bowl, lemon, banana and spatula
[6]Textures: cow, crocodile, elephant, leopard, snake, tiger and zebra

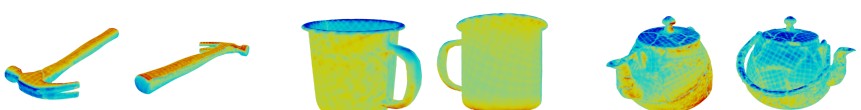

Figure 12: Model sensitivity to pose. The heatmaps denote the accuracy of the model in predicting the correct label, conditioned on a specific part of the object being visible in the image. Here, red and blue denotes high and low accuracy respectively.

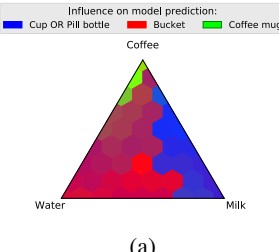
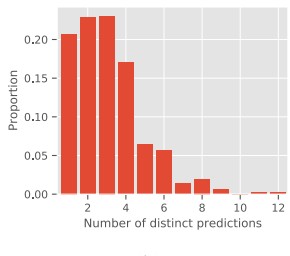
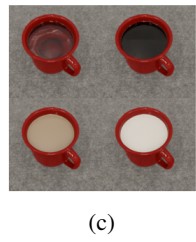

(a)

(b)

(c)

Figure 13: Testing classifier sensitivity to context: Figure (a) shows the correlation of the liquid mixture in the mug on the prediction of the model, averaged over random viewpoints (see Figure 20b for the raw frequencies). Figure (b) shows that for a fixed viewpoint, model predictions are unstable with respect to the liquid. Figure (c) shows examples of rendered liquids (water, black coffee, milk, and mixtures).

gated further through natural data collection, or—as we discuss in the upcoming section—through additional experimentation with *3DB*.

### 3.4 Case study: using *3DB* to dive deeper

Our heatmap analysis in the previous section (cf. Figure 12) showed that classification accuracy for the mug decreases when its interior is visible. What could be causing this effect? One hypothesis is that in the ImageNet training set, objects are captured in context, and thus ImageNet-trained classifiers rely on this context to make decisions. Inspecting the ImageNet dataset, we notice that coffee mugs in context usually contain coffee. Thus, the aforementioned hypothesis would suggest that the model relies, at least partially, on the contents of the mug to correctly classify it. *Can we leverage 3DB to confirm or refute this hypothesis?*

To test this, we implement a custom control that can render a liquid inside the "coffee mug" model. Specifically, this control takes water:milk:coffee ratios as parameters, then uses a parametric Blender shader (cf. Appendix G) to render a corresponding mixture of the liquids into the mug. We used the pre-packaged grid search policy, (programmatically) restricting the search space to viewpoints from which the interior of the mug was visible.

The results of the experiment are shown in Figure 13. It turns out that the model is indeed sensitive to changes in liquid, supporting our hypothesis: model predictions stayed constant (over all liquids) for only 20.7% of the rendered viewpoints (cf. Figure 13b). The *3DB* experiment provides further support for the hypothesis when we look at the correlation between the liquid mixture and the predicted class: Figure 13a visualizes this correlation in a normalized heatmap (for the unnormalized version, see Figure 20b in the Appendix G). We find that the model is most likely to predict "coffee mug" when coffee is added to the interior (unsurprisingly); as the coffee is mixed with water or milk, the predicted label distribution shifts towards "bucket" and "cup" or "pill bottle," respectively. Overall, our experiment suggests that current ResNet-18 classifiers are indeed sensitive to object context—in this case, the fluid composition of the mug interior. More broadly, this illustration highlights how a system designer can quickly go from hypothesis to empirical verification with minimal effort using *3DB*. (In fact, going from the hypothesis to Figure 13 took less than a day of work for one author.)

## 4 Physical realism

The previous sections have demonstrated various ways in which we can use *3DB* to obtain insights into model behavior in simulation. Our overarching goal, however, is to understand when models will fail in the physical world. Thus, we would like for the insights extracted by *3DB* to correspond to naturally-arising model behavior, and not just artifacts of the simulation itself. To this end, we now test the *physical realism* of *3DB*: can we understand model performance (and uncover vulnerabilities) on real photos using only a high-fidelity simulation?

To answer this question, we collected a set of physical objects corresponding to 3D models, and set up a physical room with a corresponding 3D environment. We used *3DB* to identify strong points and vulnerabilities of an ImageNet classifier in this environment, mirroring our methodology from

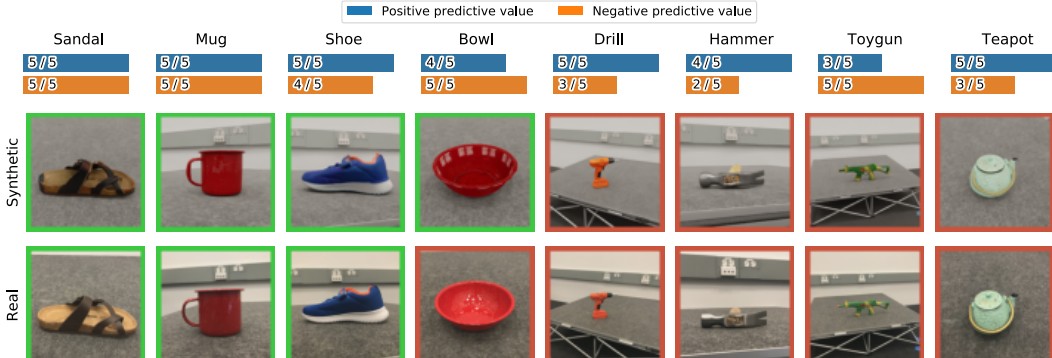

Figure 14: **(Top)** Agreement, in terms of model correctness, between model predictions within *3DB* and model predictions in the real world. For each object, we selected five rendered scenes found by *3DB* that were misclassified in simulation, and five that were correctly classified; we recreated and deployed the model on each scene in the physical world. The *positive (resp., negative) predictive value* is rate at which correctly (resp. incorrectly) classified examples in simulation were also correctly (resp., incorrectly) classified in the physical world. **(Bottom)** Comparison between example simulated scenes generated by *3DB* (first row) and their recreated physical counterparts (second row). Border color indicates whether the model was correct on this specific image.

Section 3. We recreated each scenario found by *3DB* in the physical room, and took photographs that matched the simulation as closely as possible. Finally, we evaluated the physical realism of *3DB* by comparing models' performance on the photos to what *3DB* predicted.

**Setup.** We used a studio room shown in Appendix Figure 18b for which we obtained a fairly accurate 3D model (cf. Appendix Figure 18a). We leverage the YCB [13] dataset to guide our selection of real-world objects, for which 3D models are available. We supplement these by sourcing additional objects (from amazon.com) and using a 3D scanner to obtain corresponding meshes.

We next used *3DB* to analyze the performance of a pre-trained ImageNet ResNet-18 on the collected objects in simulation, varying over a set of realistic object poses, locations, and orientations. For each object, we selected 10 rendered situations: five where the model made the correct prediction, and five where the model predicted incorrectly. We then tried to recreate each rendering in the physical world. First we roughly placed the main object in the location and orientation specified in the rendering, then we used a custom-built iOS application (see Appendix D) to more precisely match the rendering with the physical setup.

**Results.** Figure 14 visualizes a few samples of renderings with their recreated physical counterparts, annotated with model correctness. Overall, we found a 85% agreement rate between the model's correctness on the real photos and the synthetic renderings—agreement rates per class are shown in Figure 14. Thus, despite imperfections in our physical reconstructions, the vulnerabilities identified by *3DB* turned out to be physically realizable vulnerabilities (and conversely, the positive examples found by *3DB* are usually also classified correctly in the real world). We found that objects with simpler/non-metallic materials (e.g., the bowl, mug, and sandal) tended to be more reliable than metallic objects such as the hammer and drill. It is thus possible that more precise texture tuning of 3D models object could increase agreement further (although a more comprehensive study would be needed to verify this).

# 5   Related work

In this section, we give a brief overview of existing work in robustness, interpretability, and simulation that provide the context for our work. We refer the reader to Appendix A for a detailed discussion of prior work.

**Model Robustness.** The brittleness of current ML models has drawn the attention to analyze the robustness and reliability of these models. A long line of research focus on analyzing model robustness to adversarial examples [61, 20, 70, 21, 20, 12, 5, 68, 47, 44]. Another line of research involves

analyzing robustness to non-adversarial corruptions [18, 25, 32, 38, 74, 69, 23, 52]. A more closely related line of research to ours analyzes the impact of factors such as object pose and geometry by applying synthetic perturbations in three-dimensional space [28, 57, 29, 3, 35].

**Interpretability and model debugging.** *3DB* can be cast as a method for *debugging* vision models that provides users fine-grained control over the rendered scenes and thus enables them to find specific modes of failure (cf. Sections 3 and 4). Model debugging is also a common goal in intepretability, where methods generally seek to provide justification for model decisions based on either local features (e.g., saliency maps) [58, 14, 60, 50, 22, 74, 27] or global ones (i.e., general biases of the model) [7, 41, 71, 63].

**Simulated environments.** Finally, there has been a long line of work on developing simulation platforms as a source of additional training data [11, 8, 36, 73, 15, 31, 51, 55, 59, 17, 42, 64, 48, 65, 66, 67, 54]. *3DB* shares some components with many of these works (e.g., a rendering engine), but has a very different goal and set of applications, i.e., diagnosing specific failures in existing models.

## 6  Conclusion

In this work, we introduced *3DB*, a unified framework for diagnosing failure modes in vision models based on high-fidelity rendering. We demonstrate the utility of *3DB* by applying it to a number of model debugging use cases—such as understanding classifier sensitivities to realistic scene and object perturbations, and discovering model biases. Further, we show that the debugging analysis done using *3DB* in simulation is actually predictive of model behavior in the physical world. Finally, we note that *3DB* was designed with extensibility as a priority; we encourage the community to build upon the framework so as to uncover new insights into the vulnerabilities of vision models.

**Limitations.** One limitation of *3DB* is the need for high-quality 3D models for objects of interest in order to achieve photorealistic images. This requires 3D model artists and/or effective photogrammetry techniques. Additionally, creating fully realistic scenes may require more complexity than just combining a single object with a background, which is what we focus on in this paper. *3DB* does support multiple objects, and the user can programmatically specify how different objects are located relative to each other; we hope to explore this more in the future.

## Acknowledgements

Work supported in part by the NSF grants CCF-1553428 and CNS-1815221, the Google PhD Fellowship, the Open Philanthropy Project AI Fellowship, the NDSEG PhD Fellowship, and the Microsoft Corporation. This material is based upon work supported by the Defense Advanced Research Projects Agency (DARPA) under Contract No. HR001120C0015.

Research was sponsored by the United States Air Force Research Laboratory and the United States Air Force Artificial Intelligence Accelerator and was accomplished under Cooperative Agreement Number FA8750-19-2-1000. The views and conclusions contained in this document are those of the authors and should not be interpreted as representing the official policies, either expressed or implied, of the United States Air Force or the U.S. Government. The U.S. Government is authorized to reproduce and distribute reprints for Government purposes notwithstanding any copyright notation herein.

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
