# A   Detailed related work

*3DB* builds on a growing body of work that looks beyond accuracy-based benchmarks in order to understand the *robustness* of modern computer vision models and their failure modes. In particular, our goal is to provide a unified framework for reproducing these studies and for conducting new analyses. In this section, we discuss the existing research in robustness, interpretability, and simulation that provide the context for our work.

**Adversarial robustness.** Several recent works propose analyzing model robustness by crafting adversarial, i.e., *worst-case*, inputs. For example, [61] discovered that a carefully chosen but imperceptible perturbation suffices to change classifier predictions on virtually any natural input. Subsequently, the study of such "adversarial examples" has extended far beyond the domain of image classification: e.g., recent works have studied worst-case inputs for object detection and image segmentation [20, 70, 21]; generative models [43]; and reinforcement learning [34]. More closely related to our work are studies focused on three-dimensional or physical-world adversarial examples [20, 12, 5, 68, 47]. These studies typically use differentiable rendering and perturb object texture, geometry, or lighting to induce misclassification. Alternatively, Li, Schmidt, and Kolter [44] modify the camera itself via an adversarial camera lens that consistently cause models to misclassify inputs.

In our work, we have primarily focused on using non-differentiable but high-fidelity rendering to analyze a more *average-case* notion of model robustness to semantic properties such as object orientation or image backgrounds. Nevertheless, the extensibility of *3DB* means that users can reproduce such studies (by swapping out the Blender rendering module for a differentiable renderer, writing a custom control, and designing a custom search policy) and use our framework to attain a more realistic understanding of the worst-case robustness of vision models.

**Robustness to synthetic perturbations.** Another popular approach to analyzing model robustness involves applying transformations to natural images and measuring the resultant changes in model predictions. For example, Engstrom et al. [18] measure robustness to image rotations and translations; Geirhos et al. [25] study robustness to style transfer (i.e., texture perturbations); and a number of works have studied robustness to common corruptions [32, 38], changes in image backgrounds [74, 69], Gaussian noise [23], and object occlusions [52], among other transformations.

A more closely related approach to ours analyzes the impact of factors such as object pose and geometry by applying synthetic perturbations in three-dimensional space [28, 57, 29, 3]. For example, Hamdi and Ghanem [28] and Jain et al. [35] use a neural mesh renderer [40] and Redner [45], respectively, to render images to analyze the failure modes of vision models. Alcorn et al. [3] present a system for discovering neural networks' failure modes as a function of object orientation, zoom, and (two-dimensional) background and perform a thorough study on the impact of these factors on model decisions.

*3DB* draws inspiration from the studies listed above and tries to provide a unified framework for detecting *arbitrary* model failure modes. For example, our framework provides explicit mechanisms for users to make custom controls and custom search strategies, and includes built-in controls designed to range across many possible failure modes encompassing nearly all of the aforementioned studies (cf. Section 3). Users can also *compose* different transformations in *3DB* to get an even more fine-grained understanding of model robustness.

**Other types of robustness.** An oft-studied but less related branch of robustness research tests model performance on unaltered images from distributions that are close to but not identical to that of the training set. Examples of such investigations include studies of newly collected datasets such as ImageNet-v2 [**taori2020measuring**, 49, 19], ObjectNet [6], and others (e.g., [33, 56]). In a similar vein, Torralba and Efros [62] study model performance when trained on one standard dataset and tested on another. We omit a detailed discussion of these works since *3DB* is synthetic by nature (and thus less photorealistic than the aforementioned studies). As shown in Section 4, however, *3DB* is indeed realistic enough to be indicative of real-world performance.

**Interpretability, counterfactuals, and model debugging.** *3DB* can be cast as a method for *debugging* vision models that provides users fine-grained control over the rendered scenes and thus enables them to find specific modes of failure (cf. Sections 3 and 4). Model debugging is also a common goal in intepretability research, where methods generally seek to provide justification for model decisions based on either local features (i.e., specific to the image at hand) or global ones (i.e., general

biases of the model). Local explanation methods, including saliency maps [58, 14, 60], surrogate models such as LIME [50], and counterfactual image pairs [22, 74, 27], can provide insight into specific model decisions but can also be fragile [26, 4] or misleading with respect to global model behaviour [1, 60, 2, 46]. Global interpretability methods include concept-based explanations [7, 41, 71, 63] (though such explanations can often lack causal links to the features models actually use [27]), but also encompass many of the robustness studies highlighted earlier in this section, which can be cast as uncovering global biases of vision models.

**Simulated environments and training data.** Finally, there has been a long line of work on developing simulation platforms that can serve as both a source of additional (synthetic) training data, and as a proxy for real-world experimentation. Such simulation environments are thus increasingly playing a role in fields such as computer vision, robotics, and reinforcement learning (RL). For instance, OpenAI Gym [11] and DeepMind Lab [8] provide simulated RL training environments with a fleet of control tasks. Other frameworks such as UnityML [36] and RoboSuite [73] were subsequently developed to cater to more complex agent behavior.

In computer vision, the Blender rendering engine [10] has been used to generate synthetic training data through projects such as BlenderProc [15] and BlendTorch [31]. Similarly, HyperSim [51] is a photorealistic synthetic dataset focused on multimodal scene understanding. Another line of work learns optimal simulation parameters for synthetic data generation according to user-defined objectives, such as minimizing the distribution gap between train and test environments [39, 16, 9]. Simulators such as AirSim [55], FlightMare [59], and CARLA [17] (built on top of video game engines Unreal Engine and Unity) allow for the collection of synthetic training data for perception and control. In robotics, simulators include environments that model typical household layouts for robot navigation [42, 64, 48], interactive ones with objects that can be actuated [65, 66, 67], and those that include support for tasks such as question answering and instruction following [54].

While some of these platforms may share components with *3DB* (e.g., the physics engine, photorealistic rendering), they do not share the same goals as *3DB*, i.e., diagnosing specific failures in existing models.

## B  Implementation and scalability

In this section, we briefly describe the underlying architecture of *3DB*, and verify that the system can effectively scale to distributed compute infrastructure.

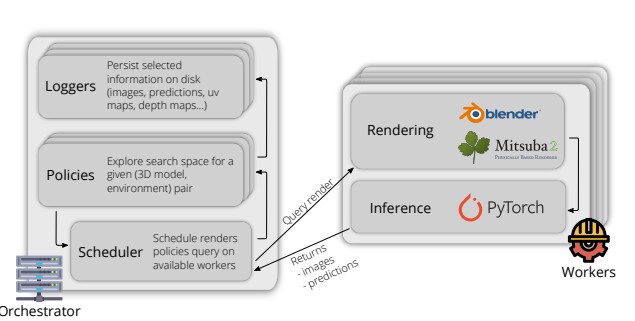
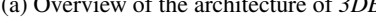
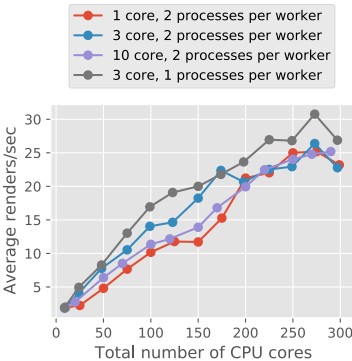

(a) Overview of the architecture of *3DB*

(b) Performance of *3DB* on a simple experiment in function of the number of CPU cores recruited for renders.

**Architecture.** To ensure scalability of this pipeline, we implement *3DB* as a client-server application (cf. Figure 15a). The main "orchestrator" thread constructs a search space by composing the user's specified controls, then uses the (user-specified) policy to find the exact set of 3D configurations that need to be rendered and analyzed. It then schedules these configurations across a set of worker nodes, whose job is to receive configurations, render them, run inference using the user's pretrained vision model, and send the results back to the orchestrator node. The results are aggregated and

written to disk by a logging module. The dashboard is implemented as a separate entity that reads the log files and produces a user-friendly web interface for understanding the *3DB* results.

**Scalability.** As discussed in Section 2, in order to perform photo-realistic rendering at scale, *3DB* must be able to leverage many machines (CPU cores) in parallel. *3DB* is designed to allow for this. It can accommodate as many rendering clients as the user can afford and the rendering efficiency of *3DB* largely scales linearly with available CPU cores (cf. Figure 15b). Note that the although the user can add as much rendering clients as they want, the number of actually used clients by the orchestrator is limited by its number of policy instances. In our paper, we run a limited number instances of policies (one instance per (env, 3D model) pair) concurrently to keep the memory of the orchestrator under control. This limits the scalability of the system as the maximum of renders that has to be done at any point in time scales with the number of policies of the orchestrator. Yet, were able to reach 415 FPS average/800 FPS peak throughput with dummy workers (no rendering), and around 100 FPS for the main experiments of this paper (e.g. physical realism experiment) which uses a complex background environment requiring substantial amount of rendering time (15 secs per image).

## C Experiment dashboard

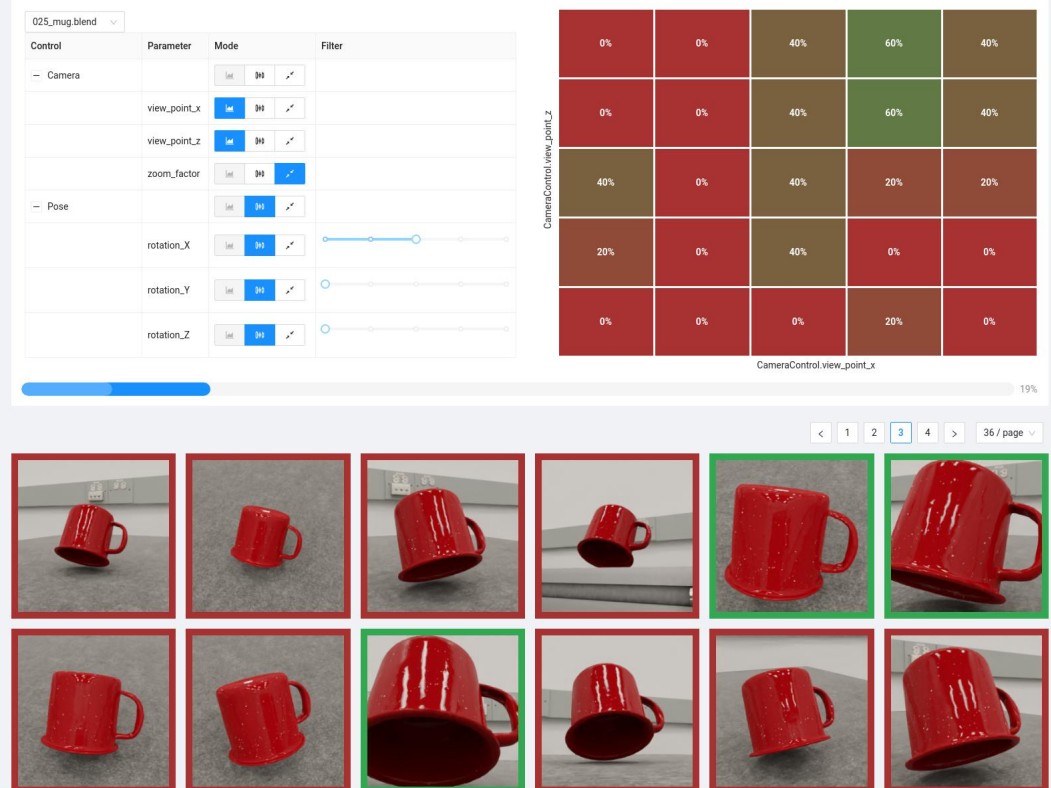

Figure 16: Screenshot of the dashboard used for data exploration.

Since experiments usually produce large amounts of data that can be hard to get a sense of, we created a data visualization dashboard. Given a folder containing the JSON logs of a job, it offers a user interface to explore the influence of the controls.

For each parameter of each control, we can pick one out three mode:

- **Heat map axis**: This control will be used as the x or y axis of the heat map. Exactly two controls should be assigned to this mode to enable the visualization. Hovering on cells of the heat map will filter all samples falling in that region.
- **Slider**: This mode enables a slider that is used to only select the samples that match exactly this particular value.
- **Aggregate**: do not filter samples based on this parameter

## D iPhone App

We developed a native iOS app to help align objects in the physical experiment (Section 4). The app allows the user to enter one or more rendering IDs (corresponding to scenes rendered by *3DB*); the app then brings up a camera with a translucent overlay of either the scene or an edge-filtered version of the scene (cf. Figure 17). We used the app to align the physical object and environment with their intended place in the rendered scene. The app connects to the same backend serving the experiment dashboard.

## E Controls

*3DB* takes an object-centric perspective, where an object of interest is spawned on a desired background. The scene mainly consists of the object and a camera. The controls in our pipeline affect this

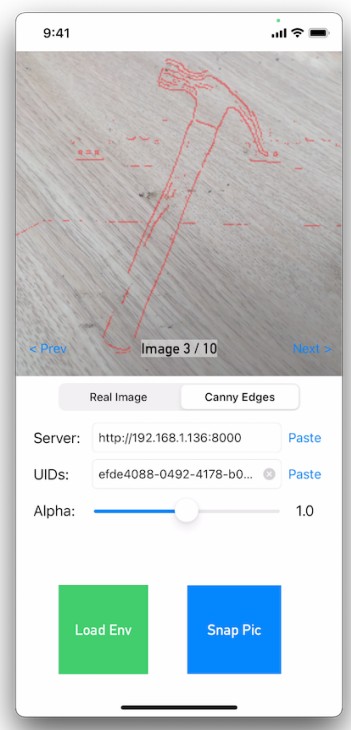

Figure 17: A screenshot of the iOS app used to align objects for the physical-world experiment. After starting the dashboard server, the user can specify the server location as well as a set of rendering IDs. The corresponding renderings will be displayed over a camera view, allowing the user to correctly position the object in the frame. The user can adjust the object transparency, and can toggle between overlaying the full rendering and overlaying just the edges (shown here).

interplay between the scene components through various combinations of properties, which subsequently creates a wide variety of rendered images. The controls are implemented using the Blender Python API 'bpy' that exposes an easy to use framework for controlling Blender. 'bpy' primarily exposes a scene context variable, which contains references to the properties of the components such as objects and the camera; thus allowing for easy modification.

*3DB* comes with several predefined controls that are ready to use. Nevertheless, users are able (and encouraged) to implement custom controls for their use-cases.

# F   Additional experimental details

For all experiments we used the pre-trained ImageNet ResNet-18 included in `torchvision`. In this section we will describe, for each experiment the specific 3D-models and environments used by *3DB* to generate the results.

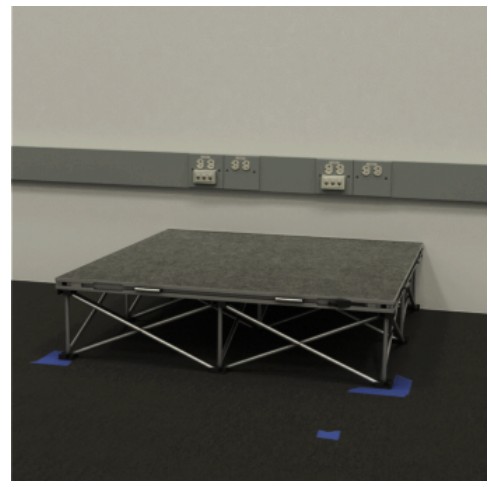

(a) Synthetic

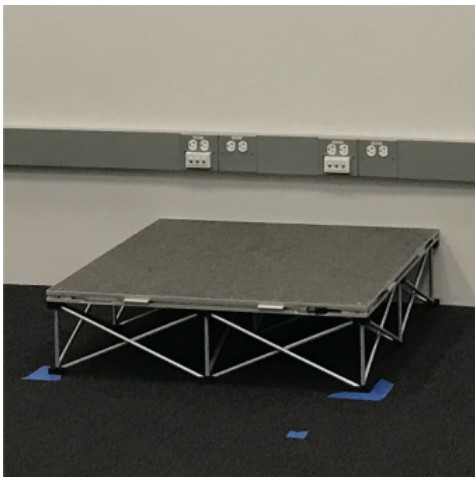

(b) Real picture (iPhone 12 Pro)

Figure 18: Studio used for the real-world experiments (Section 4).

## F.1   Sensitivity to image backgrounds (Section 3.1)

### F.1.1   Analysing a subset of backgrdounds

**Models:** We collected 19 3D-models in total. On top of the models shown on figure 21, we used models for: (1) an orange, (2) two different toy power drills, (3) a baseball ball, (4) a tennis ball, (5) a golf ball, (6) a running shoe, (7) a sandal and (8) a toy gun. Some of these models are from YCB [13] and the rest are purchased from `amazon.com` and then put through a 3D scanner to get corresponding meshes.

**Environments:** We sourced 20 2k HDRI from the website https://hdrihaven.com. In particular we used: `abandoned_workshop`, `adams_place_bridge`, `altanka`, `aristea_wreck`,
`bush_restaurant`, `cabin`, `derelict_overpass`, `dusseldorf_bridge`,
`factory_yard`, `gray_pier`, `greenwich_park_03`, `kiara_7_late-afternoon`,
`kloppenheim_06`, `rathaus`, `roofless_ruins`, `secluded_beach`,
`small_hangar_02`, `stadium_01`, `studio_small_02`, `studio_small_04`.

### F.1.2   Analyzing all backgrounds with the "coffee mug" model.

**Models:** We used a single model: the coffee mug, in order to keep computational resources under control.

**Environments:** We used 408 HDRIs from https://hdrihaven.com/ with a 2K resolution.

## F.2   Texture-shape bias (section 3.2)

**Textures:** To replace the original materials, we collected 7 textures on the internet and we modified them to make them seamlessly tilable. These textures are shown on Figure 21.

**Models:** We used all models that are shown on Figure 21.

**Environments:** We used the virtual studio environment (Figure 18).

### F.3 Orientation and scale dependence (Section 3.3)

We use the same models and environments that are used in Appendix F.1.1.

### F.4 3D models Heatmaps (Figure 12)

**Models:** For this experiment we used the set of models shown on Figure 21.

**Environments:** We used the virtual studio environment (see Figure 18).

### F.5 Case study: using 3DB to dive deeper (Section 3.4)

**Models:** We only used the mug since this experiment is mug specific.

**Environments:** We used the sudio set shown on Figure 18.

### F.6 Physical realism (Section 4)

**Real-world pictures:** All images were taken with an handheld Apple iPhone 12 Pro. To help us align the shots we used the application described in appendix D.

**Models:** We used the models shown in Figure 14.

**Environments:** The environment shown on Figure 18 was especially designed for this experiment. The goal was to have an environment that matches our studio as closely as possible. The geometry and materials were carefully reproduced using reference pictures. The lighting was reproduce through a high resolution HDRI map.

### F.7 Performance scaling (Appendix B)

The only relevant details for this experiment are the fact that we ran 10 policies (at most 5 concurrently). Each policy consisted of 1000 renders using a 2k HDRI as environment.

# G  Omitted figures

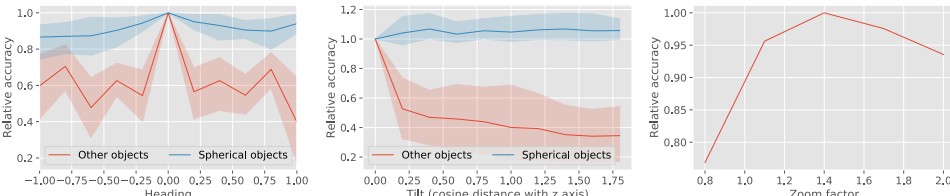

Figure 19: Additonal plots to Figure 11. We plot the distribution of model accuracy as a function of object heading (*top*), tilt (*middle*) and zoom (*bottom*), aggregated over variations in controls. For heading and tilt, we separately evaluate accuracy for (non-)spherical objects. Notice how the performance of the model degrades for non-spherical objects as the heading/tilt changes, but not for spherical objects. Also notice how the performance depends on the zoom level of the camera (how large the object is in the frame).

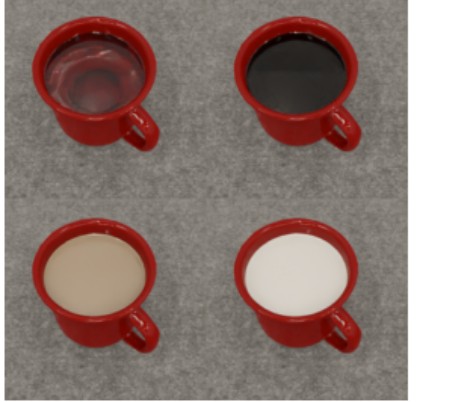

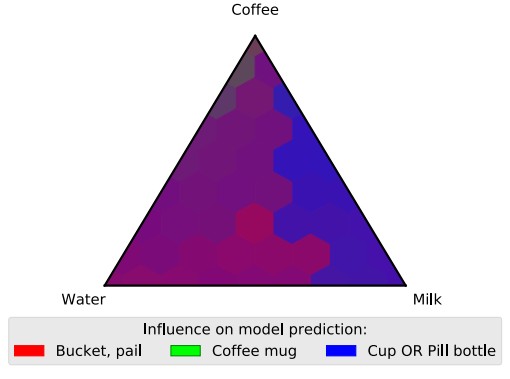

(a) Sample of the images rendered for the experiment presented in section 3.4.

(b) Un-normalized version of Figure 13-(b).

Figure 20: Additional illustration for the mug liquid experiment of Figure 13. This figure shows the correlation of the liquid mixture in the mug on the prediction of the model, averaged over random viewpoints

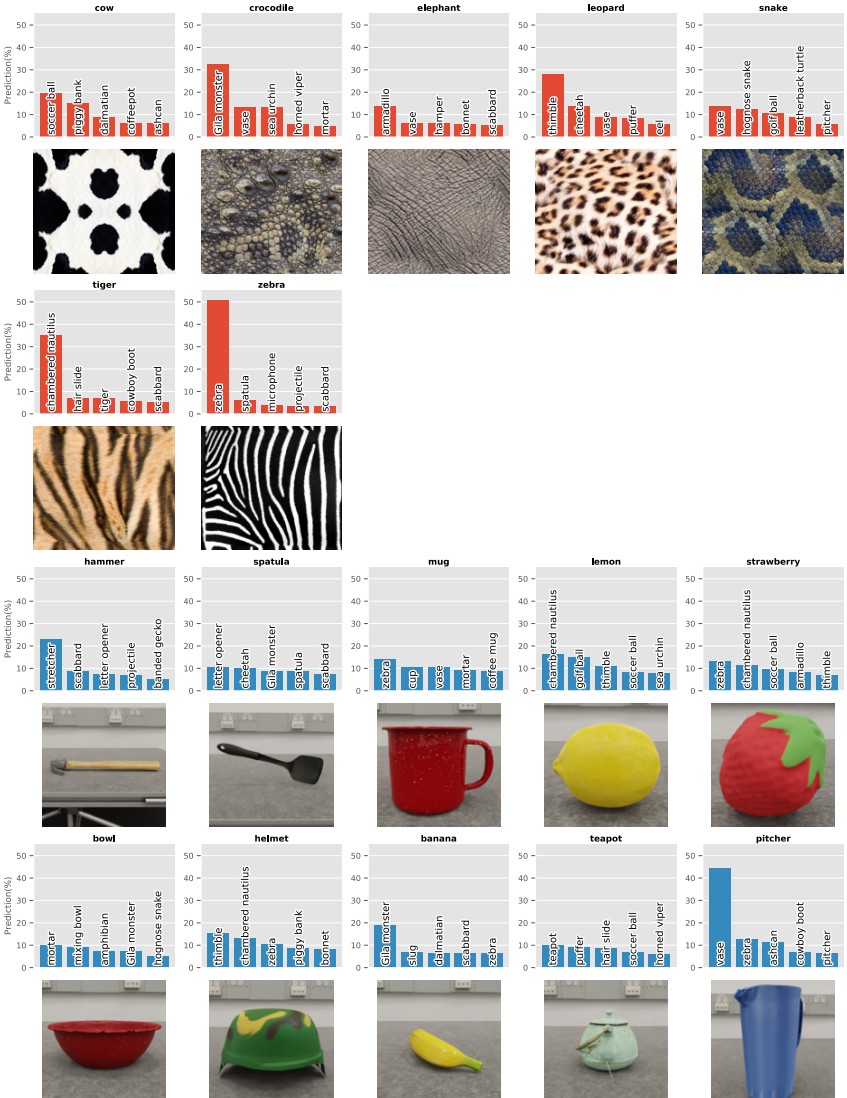

Figure 21: Additional experiment for Section 3.2. Distribution of classifier predictions after the texture of the 3D object model is altered. In the top rows, we visualize the most frequently predicted classes for each texture (averaged over all objects). In the bottom rows, we visualize the most frequently predicted classes for each object (averaged over all textures). We find that the model tends to predict based on the texture more often than based on the object.