# OpenReview forum: "3DB: A Framework for Debugging Computer Vision Models"
_NeurIPS.cc/2022/Conference — NeurIPS 2022 Accept_

### Official Review · Reviewer_Go77 · 2022-07-10

**Rating:** 6
**Confidence:** 4
**Soundness:** 3 good
**Presentation:** 3 good
**Contribution:** 3 good

**Summary:**

The paper presents 3DB — library/software package for studying computer vision models sensitivity to various transformations, from the geometrical, like camera pose change, to semantic, e.g. “Does color of the liquid in the cup influence the classification output of the model?”.
3DR is implemented as (wrapper + UI) around (Blender + deep learning model runner). It (supposedly) comes with some 3D models and background — graphical “assets”.
The paper showcases the library with several case studies — background influence, liquid color influence, and zoom/rotation influence on the classification model output.

**Questions:**

I actually, have no idea, how to evaluate such paper/tool w.r.t. NeurIPS acceptance. My gut feeling is the following (feel free to correct me if I am wrong, both AC, Rs and authors):

if the tool would be useful for machine learning community, the paper should be accepted. If not, then not.

Questions (again, non-standard)

1. How are you going to support and maintain 3DB? Would it be someone supporting it at least part time? Open source community, authors themselves/etc?
2. Do you plan any tutorial (both like web-tutorial, and like CVPR/NeurIPS tutorial), showing how to install and use 3DB?
3. Do you plan to release pre-rendered datasets, so people could run their evaluations without installing 3DB?
4. Could you please elaborate on how much time each experiment described in the paper take? With (if possible) a time breakdown: "search for 3D models online" - X hours, debugging - Y hours, running rendering & inference - Z hours, etc.


****
After-rebuttal update:

Authors have been addressed 3/4 of my questions and additionally announced to release a dataset of 5k objects under CC license. This significantly reduces my concerns. The last, not addressed one, is maintenance of the library, but let's hope for the best.
I am increasing my score to weak accept.




**Limitations:**

Yes, authors clearly stated the limitations of their framework

**Strengths And Weaknesses:**

The paper is not a standard NeurIPS paper, so this review would be a bit non-standard as well. The main contribution is not the presented experiments themselves, but rather the software package usefulness for researchers.
The package is provided in the supplementary material, so I have tried to install/run it.
Unfortunately, it is not possible, because there no “setup.py” file, also the requirements.txt are empty.

In order to try the run the library, I have looked at github for 3db, and luckily found it, under the same org name “3db”. Also, I have seen some github handles of the authors, but none of them is known to me, so I consider myself still in double blind mode.
However, if this is not the case, feel free to ignore my review and ban me.

Anyway, let me continue. The repo was more full and contained instructions how to install the thing, however, MacOS was not supported. Luckily, there was a docker option. Unluckily, it was not working either. I was able to proceed a little bit more, by fixing a couple of mistakes there (e.g. python==3.7, not python==3.7.7 for conda env), but finally gave up, as I spend on this around 1.5 hours without much success. Some paths are hardcoded as well (e.g. threedb_master) , which does not help either.

Now, I fully realize, that I probably was not motivated enough to make it work, also my devops skills definitely suck.
Let’s meet on the middle ground and say that library is not ready/documented enough for usage by beginners. Expert should definitely be able to run the 3DB, and the middle person - depending on time and motivation.


Now, when I fail to try the package myself, I fall back to evaluating it from the paper only.

Supposed UX is the following:
1. User comes up with hypothesis to test, e.g. if the chair detection performance is heavily dependent on  are
2. User creates/buys 3D model of the chair, Blender-compatible, and some backgrounds
3. User writes down the config in yaml for rendering parameters
4. The rendering and inference occurs in a parallelized way (map)
5. Results are them reduced to the visualization server, where one could generate summarization graphs, as well, as see the failure cases themselves.

This seems as attractive process except stage 2.

Weaknesses:

1. The installation process is not user-friendly
2. (stated in paper itself) you are as good, as 3D models you own. No 3D models - no 3db
3. Although, that title says “Debugging Computer Vision Models”, in fact, only classification and detection is really supported now. Tracking, matching, etc are beyond the score of 3db (at least now. One can extend the framework probably).


Note, that I don't evaluate experiments presented in the paper themselves, as they were used just as a showcase for the tool. I like the experiments, though.

---

> ### Author Response · Authors · 2022-08-02
> **Response to Reviewer Go77**
>
> We thank the reviewer for their comments. We address their specific concerns below:
>
> **[How are you going to support and maintain 3DB?]**
> We will leverage the open source community to suggest improvements to the framework.
>
> **[Do you plan any tutorial (both like web-tutorial, and like CVPR/NeurIPS tutorial), showing how to install and use 3DB?]**
> We didn’t originally but this is a good suggestion.
>
> **[Do you plan to release pre-rendered datasets, so people could run their evaluations without installing 3DB?]**
> We have multiple pre-rendered datasets available (some with more than 1M renders). However since renders are relatively inexpensive on standard CPU clusters and 3DB offers almost impossible freedom we suspect that users will want to generate their own. Furthermore, to reduce the barrier of entry we collected and plan to release a dataset of more than 5000 3D models compatible with 3DB. These models come with a Creative Commons license and cover many common household objects.
>
> **[How much time each experiment described in the paper takes?]**
> For all the experiments in this paper we used the standard YCB dataset and some custom scanned models that we bought from amazon (as described in Section 4 and Appendix E). In both cases, each model had to be cleaned ~30 min. To reduce this time in the future we will release the above-mentioned dataset of 3D models.
> Inference time was negligible. Rendering mostly depends on how many cpus are available but since most image classifiers work on small images we often reach throughputs in excess of 100 images per second. The lengthiest part of the process was the post processing and chart generation. For example generating the heatmaps took about 4h of development.
> Finally, we would like to note that it is really easy to use 3DB and go from ideas to results (once 3D models are available). In fact, going from the initial hypothesis of the case study in Section 3.4 to Figure 13 took less than a single day of work for one author.

---

> > ### Comment · Reviewer_Go77 · 2022-08-03
> > **Open source community is not a solution**
> >
> > >We will leverage the open source community to suggest improvements to the framework.
> >
> > Well, suggesting improvements is not a maintenance. I am talking about not only adding new features, but also finding and fixing bugs, making changes against API changes in other libraries (bitrot, https://en.wikipedia.org/wiki/Software_rot), etc.
> > From my (limited) experience, it is either your employer, that sponsors the project, by allowing allocating your work time, or some corporation/foundation supporting with money. Or you are doing yourselves, which is also OK, but not easy at all.

---

> > ### Comment · Reviewer_Go77 · 2022-08-06
> > **Error: Field missing:**
> >
> > I will update my review later, but want to give you update now:
> >
> > 1) I am still very sceptical about usability (and ease of use) of 3DB, and how you are going to do a maintenance, given lack of updates so far. However, I wish to be wrong.
> > 2) "Furthermore, to reduce the barrier of entry we collected and plan to release a dataset of more than 5000 3D models compatible with 3DB. These models come with a Creative Commons license and cover many common household objects." Sounds good.
> >
> >
> > Overall, I will increase my rating to weak accept.

---

> > > ### Author Response · Authors · 2022-08-08
> > > **Reply to reviewer Go77**
> > >
> > > We thank the reviewer for raising their score and all their detailed feedback. We will maintain 3DB with our current team. As we start getting feedback from users and this project gets traction, we will assign a dedicated person for maintaining it (with the help of the open-source community). We cannot disclose more details of the maintenance plan for anonymity reasons, but are committed to ensuring that 3DB remains a useful tool for the community!

---

### Official Review · Reviewer_8XFw · 2022-07-11

**Rating:** 5
**Confidence:** 3
**Soundness:** 3 good
**Presentation:** 3 good
**Contribution:** 2 fair

**Summary:**

The paper introduces 3DB, a new framework to test vision models using various renderings of 3D simulation. It allows users to discover model vulnerabilities such as textures, corruptions, geometric transformations, different backgrounds etc. Finally, a realistic benchmark is constructed based on available 3D models, to demonstrate the agreement rate between synthetic renderings and real-world images.

**Questions:**

My primary concerns are:

- Are you using YCB dataset scanning 3D objects yourselves for all 3D models? I think it only has limited tabletop objects. Is it feasible to test on more categories?

- Do you have any results (even qualitatively) on benchmarks besides image classification?

**Limitations:**

Limitations are discussed in the paper.

**Strengths And Weaknesses:**

Strengths:

- The approach is neat. Considering it's very hard to collect real-world images of the same object under different configurations, it's a good idea to utilize 3D renderings.
- One of my concern is whether the synthetic test set reflect the performance in real-world images. Failing in a synthetic image does not mean it will necessarily fail in the real world. This is resolved relatively well in Section 4 physical realism and I appreciate the experiment. Still, 40 test images look like a small test set. But I understand how hard it is to collect real-world images similar to a rendering.


Weaknesses:

- The approach to test vision models are fairly limited by avavilable 3D resources. Most vision models mentioned here only have 2D predictions -- which means the ground truth is relatively easy to collect compared with 3D models. I think it's very hard to completely test the robustness of the model. For example, it's nearly impossible to find a lot of 3D models for each ImageNet category.

- Only image classification is presented while 3DB should support more vision benchmarks such as object detection and segmentation. It is non-trivial and more challenging to render synthetic images for object detection since multiple objects are involved. Randomly assigning a 3D pose may not reflect the real-world distribution at all.

---

> ### Author Response · Authors · 2022-08-02
> **Response to Reviewer 8XFw**
>
> We thank the reviewer for their comments. We address their specific concerns below:
>
> **[Only YCB dataset?]**
> We supplement YCB objects by sourcing additional objects from Amazon, and using a 3D scanner to obtain corresponding meshes as described in Section 4 and Appendix E. The reason we don’t use only YCB objects is that, as the reviewer mentioned, it only has limited tabletop objects.
>
> **[Need for 3D models]**
> One limitation of our work is the need for 3D models, which we discuss in the limitations section of our paper. We believe though that discovering vulnerabilities in ML pipelines is an important enough problem that people would want to invest time and money to solve. Once 3D models are available (one can use photogrammetry to get started), it is straightforward to do the analysis using 3DB. In fact, going from the initial hypothesis of the case study in Section 3.4 to Figure 13 took less than a single day of work for one author.
>
> **[Results beyond image classification]**
> Indeed, we agree that object detection and semantic segmentation can be more challenging in the case where there are multiple objects in the scene. Originally, we wanted to show that users will be able to glean in-depth insights into model performance from 3DB, and thus elected to prioritize depth of analysis with a single classifier rather than breadth. However, we appreciate the reviewer’s concern about proving the robustness and extensibility of 3DB to other modalities. 3DB is built to be extensible in the sense that one can add any controls they want that allow them to manipulate objects and environments without restrictions, so 3DB certainly supports other modalities.

---

> > ### Comment · Reviewer_8XFw · 2022-08-07
> > **Re: Response to Reviewer 8XFw**
> >
> > Thanks for the explanation! I understand how difficult it is to capture multiple objects in the scene. It basically verified my initial guess. I'm happy to keep my rating.

---

### Official Review · Reviewer_8cBx · 2022-07-11

**Rating:** 6
**Confidence:** 4
**Soundness:** 3 good
**Presentation:** 3 good
**Contribution:** 3 good

**Summary:**

The paper presents a framework for systematic testing and debugging of vision models. The framework uses photorealistic simulation and considers different variables, as different object categories, HDRI backgrounds and lighting conditions, different textures, scene transformations as well as custom controls. By considering different combinations of the variables involved, the framework is quite useful in finding vulnerabilities of machine learning models with respect to specific scene configurations.

**Questions:**

1. The text also briefly discusses the possibility to use the framework for object detection tasks. It would be interesting to show some example
2. Why are Cup and Pill bottle considered together in Figure 13?

**Limitations:**

The paper openly discusses limitations. Additionally, some crucial aspects, such as the real vs simulated world domain gap, are covered in depth in the text.

**Strengths And Weaknesses:**


Originality
-----------
Simulation based methods for testing and validating computer vision systems, in general, and deep learning models in particular, have been largely considered in the past. Nevertheless, the proposed method is quite comprehensive, providing control over multiple crucial aspects, while at the same time it is highly customizable.

Quality and Clarity
-------------------
The paper is well written and easy to read. To support the proposed framework, multiple well known challenges of computer vision models are highlighted, where it is shown that significant results from the corresponding literature can be quickly and easily reproduced with the help of the proposed framework. A toy hypothesis is also considered, and it is shown that it can be easily accommodated with the help of some customization. The domain gap between real and simulated data is also discussed in sufficient detail, and quantitative comparisons are provided.

Significance
------------
The use of the proposed framework can significantly simplify the process of testing and validation of computer vision models. The framework is likely more useful for testing and validating different vision models rather than discovering novel general failure modes, as the modes of failure covered by the framework have largely being explored. Nevertheless, it is important that the framework allows for customization, as this can increase also its applicability in searching for novel controls/aspects with respect to which machine learning models may be particularly susceptible.

Minor comments
--------------
* L.122 classifier sensitivity classifiers
* The order the figures are shown does not follow the order in which they appear in the text

---

> ### Author Response · Authors · 2022-08-02
> **Response to Reviewer 8cBx**
>
> We thank the reviewer for their comments. We address their specific concerns below:
>
> **[Object detection tasks]**
> Originally, we wanted to show that users will be able to glean in-depth insights into model performance from 3DB, and thus elected to prioritize depth of analysis with a single classifier rather than breadth. However, we appreciate the reviewer’s concern about proving the robustness and extensibility of 3DB to other modalities. As mentioned in the paper, switching models other tasks ( such as objection detection) is trivial and requires no or minimal addition to the code.
>
> **[Why are Cup and Pill bottle considered together in Figure 13?]**
> This is just for visualization reasons so that we have mixtures of the major three colors, but we can also use separate colors for each of them. But really what the plot is saying is that the model predicts either Cup or pill bottle when it is filled with milk.
>
> **[Minor comments]**
> We will fix those!

---

### Official Review · Reviewer_pi9t · 2022-07-16

**Rating:** 4
**Confidence:** 4
**Soundness:** 2 fair
**Presentation:** 3 good
**Contribution:** 2 fair

**Summary:**

A software framework for testing, debugging computer vision models via use of photorealistic simulated data is described. The utility of the framework is demonstrated through a wide range of use cases and the authors illustrate how vulnerabilities in ML models for vision can be discovered. The framework allows for flexible evaluation of robustness and allows the impact of multiple factors to be studied. There is a discussion of how simulated data analysis transfers to real-world data through systematic experiments that attempt to match real to simulated settings. The software will be open-sourced.


**Questions:**

 I appreciate the level of work going into buliding a framework that can support systematic debugging and testing of computer vision models. Simulation for performance modeling has been studied in the 90's and more recently in the last decade in the context of ML and deep learning.  Can you specifically comment what is unique about your framework that allows scalable testing and validation?

How does your framework compare to frameworks that support probabilistic programming ?  These frameworks allow natural integration of graphics engines with probabilistic programs for synthesis of image datasets and exploit them in an inference loop.

Apart from the software tool and properties demonstrated, do you have specific methodological contributions for syn to real transfer/model validation of simulator?




**Limitations:**

In the last decade, a number of papers have studied the use of synthetic data for ML including some of the early papers that illustrate in detail performance characterization methodology with synthetic data and deep learning (in the context of semantic segmentation).   The authors may benefit from the series of papers below:
V. S. R. Veeravasarapu et al, Adversarially Tuned Scene Generation. CVPR 2017: 6441-6449
V. S. R. Veeravasarapu et al, Model-Driven Simulations for Computer Vision. WACV 2017: 1063-1071
V. S. R. Veeravasarapu et al, Simulations for Validation of Vision Systems. CoRR abs/1512.01030 (2015).

**Strengths And Weaknesses:**

Strengths:
A useful framework for simulated data generation, testing and debugging of computer vision models.  The experimentation is adequate to illustrate the utility of the tools.  The part involving the matching of simulated data to real data acquisitions and the experimentation with particular hypotheses on model performance (section 3.4 and 4) are interesting.

Originality:  The concepts presented and the tool itself are not novel and the results are largely confirmatory of well known behavior of ML models (as expected).

Clarity: Well written and clear.

Significance:  The paper in its present form may not yet be significant, although if scaled systematically the paper has significant potential.  The paper can benefit from a principled incorporation of performance characterization methodology articulated in the 90's in computer vision. (see haralick.org). Incorporation of probabilistic generative models (including the imaging device and rendering pipeline) and enabling of formal comparison of simulated data statistics to real data statistics can enable the tool to be of practical use.

---

> ### Author Response · Authors · 2022-08-02
> **Response to Reviewer pi9t**
>
> We thank the reviewer for their comments. We address their specific concerns below:
>
> **[What is unique about your framework that allows scalable testing and validation?]**
> What makes 3DB unique is its modular design that abstracts away all the rendering details and exposes a user-friendly interface to simulate anything the user wants with user defined policies and controls. This abstraction, combined with an automatic parallelization of renders on many worker nodes, allows for scalable testing and validation of ML models. While some of the platforms we mention in the related work section may share components with 3DB (e.g., the physics engine, photorealistic rendering), they do not share the same goals as 3DB, i.e., diagnosing specific failures in existing models, and they are not as flexible as 3db, in the sense that they allow some fixed transformations only, or require domain knowledge of how rendering works.
>
> **[Comparison to frameworks that support probabilistic programming]**
> We thank the reviewer for pointing out this connection! Although we are not familiar with the probabilistic programming literature and are unaware of any rendering engine that supports probabilistic programming, we would be happy to discuss 3DB in relation to any of these frameworks, and will do a thorough literature review and contextualization in our next revision. In general, we believe that 3DB is unique in its combination of ease of use, extensibility, and compositionality.
>
> We would also very much appreciate it if the reviewer could point us to specific frameworks they think are relevant! We would be happy to discuss these explicitly in our next revision.
>
> **[Methodological contributions for synthetic to real transfer/model validation of simulator]**
> 3DB uses blender in order to get photorealistic renders so that it effectively diagnoses failure modes in ML models. The most relevant section to the reviewer’s concern is likely Section 4, where we verified that our rendered images actually identify real failure modes by replicating the scenes in a physical setting. To run the experiments of Section 4 we built a rudimentary mobile app (which we can open-source) for properly positioning the objects, and also made use of off-the-shelf 3D scanning technology to get the models.

---

> > ### Comment · Reviewer_pi9t · 2022-08-09
> > **Rebuttal addresses some questions, but there are still gaps.**
> >
> >
> > Thank you for the detailed rebuttal.  A key reference for probabilistic programming include - Kulkarni et al, 'Picture: A probabilistic programming language for scene perception', CVPR 2015.   There are several frameworks that incorporate 3D graphics and rendering (e.g. Pyro: Deep probabilistic programming framework in 2017 from Uber ).
> >
> > I keep my rating as it reflects my current opinion on the contributions of the paper.

---

### Meta-Review · Area_Chair_KgnF · 2022-08-26

**Recommendation:** Accept
**Confidence:** Less certain

**Metareview:**

Reviewers found the presented work to be a usefulf framework, the paper to contain adequate experiments and intereting demonstrations of the framework's capabilities, and be overall well written. They appreciated tha significatn prior results can be replicated easily with the the proposed framework. On the flip side, the paper presents no new fundamentally new tools, no new results, or methodoligcal contributions.

One reviewer pointed out a missing connection to probabilistic programning frameworks and referred to several related works. Checking the referenced papers and tools:
    - Pyro is an established library, but is barely connected to any rendering at all and has a different use case than the presented paper.
    - Picture seems to never have made it beyond a Julia-based alpha stage with "under heavy initial development and has a lot of known bugs" (author's github) and is thus of no practical importance.
    - Other referenced papers have no published source code.
Probailistic programming is indeed a related direction worth discussing, but the existing tools seem to be far away from the proposed work.

Overall, the presented framework could be an interesting debugging tool for researchers and practicioners. Possibly the biggest concern targets maintenance - The value of an open source framework like the one presented here stems a lot from dedicated ongoing maintenance efforts.
I thus strongly recommend following the suggestions by reviewers and adding tutorials (ideally reproducing all the experiments in the paper) as well publishing as the collection of shapes promissed in the rebuttal.

**Award:**

No

---

### Decision · Program_Chairs · 2022-09-14

Accept